# Terpenoids from Marine Soft Coral of the Genus *Xenia* in 1977 to 2019

**DOI:** 10.3390/molecules25225386

**Published:** 2020-11-18

**Authors:** Shean-Yeaw Ng, Chin-Soon Phan, Takahiro Ishii, Takashi Kamada, Toshiyuki Hamada, Charles Santhanaraju Vairappan

**Affiliations:** 1Laboratory of Natural Products Chemistry, Institute for Tropical Biology and Conservation, Universiti Malaysia Sabah, Kota Kinabalu 88400, SBH, Malaysia; sheanyeaw@ums.edu.my (S.-Y.N.); yuna123@hotmail.my (C.-S.P.); 2Department of Bioscience and Biotechnology, Faculty of Agriculture, University of the Ryukyus, 1 Senbaru, Nishihara, Okinawa 903-0213, Japan; ishiit@agr.u-ryukyu.ac.jp; 3Department of Materials and Life Science, Faculty of Science and Technology, Shizuoka Institute of Science and Technology, 2200-2 Toyosawa, Fukuroi, Shizuoka 437-8555, Japan; takashi.kamada800@gmail.com; 4Graduate School of Science and Engineering, Kagoshima University, 1-21-35 Korimoto, Kagoshima 890-0065, Japan; thamada@sci.kagoshima-u.ac.jp

**Keywords:** marine soft corals, *Xenia* sp., natural products, terpenes, bioactivity

## Abstract

Members of the marine soft coral genus *Xenia* are rich in a diversity of diterpenes. A total of 199 terpenes consisting of 14 sesquiterpenes, 180 diterpenes, and 5 steroids have been reported to date. Xenicane diterpenes were reported to be the most common chemical skeleton biosynthesized by members of this genus. Most of the literature reported the chemical diversity of *Xenia* collected from the coral reefs in the South China Sea and the coastal waters of Taiwan. Although there was a brief review on the terpenoids of *Xenia* in 2015, the present review is a comprehensive overview of the structural diversity of secondary metabolites isolated from soft coral genus *Xenia* and their potent biological activity as reported between 1977 to 2019.

## 1. Introduction

Marine-derived terpenoids have attracted the interest of natural product chemists around the world, leading to the discovery of many novel chemical skeletons and possible lead drug candidates. In recent years, various species of marine soft corals from Taiwan and the South China Sea have been chemically investigated for their chemical diversity and potential as bioactive compounds [1,2,3,4,5,6,7,8]. Among them, marine soft coral of the genus *Xenia* (Alcyonacea, Xeniidae) has afforded numerous new terpenoids with diverse structures and few biological activities. These terpenoids are mainly dominated by xenicane-type or xenia diterpenoids. Some of these diterpenes have also been found in the alcyonacean soft corals such as *Anthelia glauca* [9], *Asterospicularia laurae* [10], *Capnella thyrsoidea* [11], *Eleutherobia aurea* [12], *Heliopora coerulea* [13], and *Sinularia gibberosa* [14] and in several gorgonian species [15,16]. In this context, a total of 199 terpenes consisting of 14 sesquiterpenes, 180 diterpenes, and 5 steroids has been reported to date. These terpenes were isolated from species of the soft coral *Xenia*, including *X. blumi*, *X. crassa*, *X. elongata*, *X. faraunensis*, *X. florida*, *X. lilielae*, *X. macrospiculata*, *X. membranacea*, *X. novae-britanniae*, *X. obscuronata*, *X. puerto-galerae*, *X. stellifera*, *X. umbellata*, *X. viridis,* and *Xenia* sp., many of which were collected at Green Island in Taiwan, regions of Kagoshima and Okinawa in Japan, Gulf of Eilat at the Red Sea, Townsville in Australia, and Sarcelle Pass south of New Caledonia. A review about *Xenia* terpenoids was reported in 2015 that listed terpenoids and their anticancer properties. However, in an effort to better understand the chemistry of marine-derived natural products from the soft coral genus *Xenia* and their bioactive potentials, this review summarizes the collection locations and chemical diversity of specimens and the biological properties of this genus that have been obtained from literature published between 1977 and 2019 (1997 to 2014 were re-reviewed). 

## 2. Diterpenes

Diterpenes are synthesized by four isoprene units, and their basic structure starts with the molecular formula C_20_H_32_. Many have a variety of modifications such as methylation, acetylation, hydroxylation, epoxidation, and oxygenation, giving them additional carbon and oxygen atoms in their molecular formula. Xenicane diterpenes contain a cyclononane skeleton. Figure 1 shows the three structurally different types of xenicane diterpenes: xenicins (possessing a 2-oxabicyclo[7.4.0]tridecane ring system) [17], xeniolides (the lactone derivatives of xenicins) [18], and xeniaphyllanes (containing a bicyclo[7.2.0]undecane ring system) [19]. Subsequently, more groups were categorized, such as floridicins (tricarbocyclic diterpenoid) [20], xeniaethers (bearing a oxabicyclo[7.3.0]undecane ring system) [21], and azamilides (having an opened A-ring with the nine-membered carbocyclic skeleton acylated with a series of C16–C20 saturated fatty acids) [22]. Since then, more novel carbon skeletons have been discovered in the 21st century such as blumiolide-A with cyclooctane system of xeniolide [23], umbellactal with γ-lactone fused to a bicyclo[4.3.0]nonane ring system [24], xenibellol A with tetracyclic system [25], and xenibellal with norditerpenoid of cyclononane system [26]. 

### 2.1. Xenia Blumi

In 2005, eight new diterpenoids, namely blumiolides A–C (**1**–**3**), 9-deoxy-isoxeniolide-A (**4**), 9-deoxy-7,8-epoxy-isoxeniolide-A (**5**), 9-deacetoxy-7,8-epoxy-13-epi-xenicin (**6**), 9-deoxy-7,8-epoxy-xeniolide-A (**7**), and blumicin-A (**8**) were isolated from *X. blumi* collected from Green Island, Taiwan [23]. Blumiolide-A was reported as a new skeleton of xeniolide with cyclooctane system, and blumiolide-C (**3**) showed potent cytotoxicity against HT-29 colorectal cancer and P-388 murine leukemia cell lines at ED_50_ 0.5 and 0.2 µg/mL, respectively (Figure 2) [23].

### 2.2. Xenia Crassa

One new diterpenoid, 9-deacetoxyxenicin (**9**), was isolated from *X. crassa* collected at John Brewer Reef, Australia [27] (Figure 2).

### 2.3. Xenia Elongata

In 1977, the first representative of the xenicins, xenicin (**10**), was isolated from *X. elongata* collected at Heron Island, Australia [17]. After 18 years, one new diterpenoid, deoxyxeniolide B (**11**) was isolated from *X. elongata* collected from Nichinan Coast, Miyazaki Prefecture, Japan [28]. This compound exhibited ichthyotoxicity against mosquito fish (*Orizias latipes*) at LC_100_ 15 ppm within 1 h. In 2007, four new diterpenoids (**12**–**15**) without given names were isolated from *X. elongata* cultured at Rutgers University [29]. Furthermore, compound **12** was found to be able to induce apoptosis in MCF 7 (breast carcinoma) cell lines. In 2014, two new diterpenoids (**16** and **17**) without given names were isolated from *X. elongata*, purchased from Ocean Reef Aquariums and cultured at Rutgers University [30]. In addition, compound **17** showed inhibition of histone deacetylase (HDAC6) with IC_50_ ~80 µM (Figure 2) [30].

### 2.4. Xenia Faraunensis

In 1994, three new diterpenoids, xeniafaraunols A and B (**18**,**19**) and faraunatin (**20**), were isolated from *X. faraunensis* collected from the Red Sea [31]. These three compounds **18**–**20** exhibited cytotoxic activities against P-388 murine leukemia cells (IC_50_ = 1.2 µg/mL) (Figure 3) [31].

### 2.5. Xenia Florida

In 1994, a new class of tricarbocyclic diterpenoid, floridicin (**21**), was isolated from *X. florida* collected at Bonotsu, Kagoshima Prefecture, Japan [20]. This was the first diterpenoid found possessing an aldehyde group in a soft coral *Xenia* sp. This was followed by the isolation of four new tricarbocyclic diterpenoids, 2-O-methylfloridicin (**22**) and floridicins A–C (**23**–**25**), from *X. florida* collected at Bonotsu, Kagoshima Prefecture, Japan [32]. In 1998, seven new diterpenoids, florlides A–E (**26**–**30**) and florethers A and B (**31** and **32**), were isolated from *X. florida* collected at Bonotsu, Kagoshima Prefecture, Japan [33]. In 2000, six new diterpenoids, namely 11-*O*-methylflorlide A (**33**), 2β-epoxyfloridicin (**34**), xeniafaraunol B (**35**), and florlides F–H (**36**–**38**), were isolated from *X. florida* collected at Bonotsu, Kagoshima Prefecture, Japan [34]. In 2005, three new diterpenoids, xeniolactones A–C (**39**–**41**), were isolated from *X. florida* collected at Green Island, off Taiwan [35]. In 2006, three new diterpenoids, florxenilides A–C (**42**–**44**), were isolated from *X. florida* collected at Green Island, off Taiwan [36]. The florxenilide B (**43**) showed strong cytotoxicity against Hepa 1–6 human liver carcinoma cells at ED_50_ 1.88 µg/mL (Figure 3) [36].

### 2.6. Xenia Lilielae

In the only report for this species, two new diterpenoids, 4,14-diepoxyxeniaphyllene (**45**) and 4,14-diepoxy-xeniaphyllenol-A (**46**), were isolated from *X. lilielae* collected from the Gulf of Eilat, the Red Sea [37] (Figure 4).

### 2.7. Xenia Macrospiculata

One year after the discovery of xenicin, a second compound of xenicin, xeniculin (**47**), was isolated from *X. macrospiculata* collected at the Gulf of Eilat, the Red Sea [38]. In the same year, two new xeniolides, xeniolides A and B (**48** and **49**), were reported from *X. macrospiculata* collected at the Gulf of Eilat, the Red Sea [18]. In 1980, xeniolide-B 9-acetate (**50**), 7,8-epoxyxeniolide-B (**51**), 4,5-epoxyxeniaphyllenol (**54**), isoxeniaphyllenol (**55**), xeniaphyllandiol 14-acetate (**58**), and 4,5-epoxy-14,15-xeniaphyllandiol (**59**) were first isolated from *X. macrospiculata* collected at the Gulf of Eilat, the Red Sea [19]. In the same year, xenialactol (**52**), xeniaphyllenol (**53**), 4,5-epoxyisoxeniaphyllenol (**56**), and 14,15-xeniaphyllandiol (**57**), were first discovered in both *X. obscuronata* from the Gulf of Suez (Red Sea) and *X. macrospiculata* from the Gulf of Eilat (Red Sea) [19]. In 1983, 4,5-epoxyxeniaphyllan-14,15-diol (**60**) was isolated from *X. macrospiculata*, *X. obscuronata*, and *X. lilielae* collected at the Gulf of Eilat, the Red Sea [37]. Xeniaphyllenols B and C (**61** and **62**) were also isolated from *X. macrospiculata* (Gulf of Eilat, the Red Sea) by the same research group [37] (Figure 4).

### 2.8. Xenia Membranacea

In 1987, two new diterpenoids, havannahine (**63**) and desoxyhavannahine (**64**), were isolated from *X. membranacea* collected at Sarcelle Pass, south of New Caledonia [39]. In 1988, four new diterpenoids, namely havannachlorhydrine-11(19) (**65**), havannachlorhydrine-7(18) (**66**), havannadichlorhydrine-7(18),11(19) (**67**), and désaceétyl-13 havannachlorhydrine-11(19) (**68**), were isolated from *X. membranacea* collected at Sarcelle Pass, south of New Caledonia [40]. To date, these compounds **65**–**68** are the only halogen-based diterpenes found in soft coral *Xenia* sp. In 1989, a pair of new diastereomeric diterpenoids, 11,19-desoxyhavannahine (**69**) and 7,8,9-epi-11,19-desoxyhavannahine (**70**), were isolated from *X. membranacea* collected at Sarcelle Pass, south of New Caledonia [41]. In 1990, 17 new diterpenoids, namely acetoxy-19-havannabol-11 (**71**), chloro-18 acetoxy-19 havannadiol-7,11 (**72**), havannabol-7 (**73**), chloro-19 havannadiol-7,11 (**74**), epi-8,9 deoxy-11,19 havannachlorhydrine-7,18 (**75**), bis-deoxy-7,18,11,19 havannahine (**76**), desacetyl-9 epoxy-7,8 epi-13 xénicine (**77**), desacetyl-9 acetoxy-18 epoxy-7,8 epi-13 xenicine (**78**), desacetyl-9 hydroxy-6 epi-13 xenicine (**79**), xenione (**80**), acetoxy-18 xenione (**81**), desacetoxy-9 epi-13 xenicine (**82**), desacetoxy-9 oxo-9 epi-13 xenicine (**83**), acide branacenoïque (**84**), branacennoate de metyhle (**85**), branacenal (**86**), and diepoxy-7(8),10(11) dichloro-18,19 isoxeniolide A (**87**), were isolated from *X. membranacea* collected at Sarcelle Pass, south of New Caledonia [42]. However, the stereochemistry of these compounds has remained undetermined to date (Figure 5).

### 2.9. Xenia Novae-Britanniae

In 1979, three new xenicin-related diterpenoids were isolated from *X. novae-britanniae* collected at Laing Island, Papua New Guinea: 13-epi-9-desacetylxenicin (**88**), isoxeniolide-A (**89**), and 7,8-oxido-isoxeniolide-A (**90**) [43]. Besides, a total of seven new xenicane-type diterpenoids, novaxenicins A–D (**91**–**94**) and xeniolides I–K (**95**–**97**), were isolated from *X. novae-britanniae* collected at Kitagamwa, southern Kenya, in 2006 [44]. Furthermore, novaxenicin B (**92**) induced apoptosis in transformed HEK-293 mammalian cells at a concentration of 1.25 µg/mL, and xeniolide I (**95**) showed antibacterial activity at a concentration of 1.25 µg/mL against *Escherichia coli* and *Bacillus subtilis* (Figure 6) [44].

### 2.10. Xenia Obscuronata

Two new diterpenoids, xenialactol-D (**98**) and xeniolide-E (**99**), were first recorded in both *X. obscuronata* from the Gulf of Suez, the Red Sea, and *X. macrospiculata* from Gulf of Eilat, the Red Sea [19]. In the same year, two new diterpenoids, 9-deacetoxy-14,15-deepoxyxeniculin (**100**) and 9-deacetoxy-14,15-deepoxyxeniculin 7,8-epoxide (**101**), were isolated from *X. obscuronata* collected at the Gulf of Suez, the Red Sea [19]. In 1983, one new diterpenoid, xeniaphyllantriol (**102**), was isolated from *X. obscuronata* collected at the Gulf of Eilat, the Red Sea [37] (Figure 6).

### 2.11. Xenia Umbellata

In 2002, the *X. umbellata*, which was collected from Green Island, Taiwan, exhibited seven new diterpenoids: 9-deoxyxeniloide-E (**103**), 9-deoxy-7,8-epoxyxeniloide-E (**104**), xeniolide-G (**105**), 9-deoxyxenialactol-C (**106**), xenibecin (**107**), xeniolide-H (**108**), and xenitacin (**109**) [45]. In addition, xeniolide-G (**105**) showed potent cytotoxicity against P-388 (obtained from Department of Medicinal Chemistry and Pharmacognosy, University of Illinois at Chicago) cell line at ED_50_ 0.04 µg/mL, while xenitacin (**109**) showed potent cytotoxicity against A-549 (purchased from American Type Culture Collection), HT-29 (purchased from American Type Culture Collection), and P-388 (obtained from Department of Medicinal Chemistry and Pharmacognosy, University of Illinois at Chicago) cell lines at ED_50_ 3.26, 1.12, and 1.09 µg/mL, respectively. In 2005, four new diterpenoids with novel skeleton, namely umbellactal (**110**), xenibellols A and B (**111** and **112**), and xenibellal (**113**), were isolated from *X. umbellata* (Green Island, off Taiwan) and showed cytotoxicity against P-388 cells with ED_50_ 3.6, 3.6, 2.8, and 3.2 µg/mL, respectively [24,25,26], as shown in Figure 6. In 2006, 11 new diterpenoids, namely umbellacins A–I (**114**–**122**), 14,15-epoxy-xeniolide H (**123**), and 3-acetyl-14,15-epoxy-xeniolide H (**124**), were isolated from *X. umbellata* collected at Green Island, off Taiwan [46]. Cytotoxicity of umbellacins B (**115**), D–F (**117**–**119**), and H–I (**121**,**122**) against P-388 cells showed ED_50_ of 1.6, 4.2, 3.8, 3.7, 3.4, and 3.6 µg/mL, respectively. In the following year, two new xeniolides, xenibelatols A and B (**125** and **126**), were isolated from *X. umbellata* collected at Green Island, off Taiwan [47]. In 2016, one new diterpenoid, xeniumbellal (**127**), was isolated from *X. umbellata* collected at Red Sea Coast, Jeddah, Saudi Arabia [48]. Moreover, xeniumbellal (**127**) and pentahydroxygorgosterol (**201**) displayed antibacterial activity against *Acinetobacter baumannii* with MIC values of 0.22 and 0.28 mM, respectively [48] (Figure 7).

### 2.12. Xenia Viridis

In 1979, the first modified tricyclic xenicin (**128**) was isolated from *X. viridis* collected at Old Reef, off Townsville, Australia [27]. In 2007, two new xenicins, 16,17-diacetoxy-9,13-deacetoxyxenicin (**129**) and 16,17-diacetoxy-7,8-oxirene-9,13-deacetoxyxenicin (**130**), were isolated from *X. viridis* collected at New Caledonia [49] (Figure 7).

### 2.13. Xenia sp.

Two pairs of new diastereomeric diterpenes, the first pair being xeniolone (**131**) and isoxeniolone (**132**) and the second pair being hydratoxeniolone (**133**) and hydratoisoxeniolone (**134**), and a new germacrane-type diterpenoid, germacrexeniolone (**135**), were isolated from *Xenia* sp. collected at Zamami-jima, Okinawa Prefecture, Japan [50,51]. This was the first isolation of perhydroazulene skeleton diterpenoids from the soft coral of *Xenia* sp. In 1995, two new diterpenoids, 9-deoxyxeniolide-A (**136**) and 9-deoxyxeniolide-B (**137**), were isolated from *Xenia* sp., and the 9-deoxyxeniolide-A (**136**) showed the greatest potency of antibacterial activity [52]. In the same year, six new diterpenoids, namely xeniatine A (**138**), xeniatine A epoxide (**139**), isoxeniatines A and B (**140** and **141**), and xeniaethers A and B (**142** and **143**), were isolated from *Xenia* sp. collected at Bonotsu, Kagoshima Prefecture, Japan [21]. These compounds, xeniaethers A and B (**142** and **143**), represented the first isolation of diterpenoids containing a tetrahydrofuran fused to a nine-membered ring from soft coral *Xenia* sp. In 1996, seven new diterpenoids, namely xeniaethers F–H (**144**–**146**), isoxeniatine C (**147**), azamials A and B (**148** and **149**), and xenicindial (**150**), were isolated from soft coral *Xenia* sp. collected at Sata Cape, Kagoshima Prefecture, Japan [53]. In the same year, the same research group reported 13 new diterpenoids, azamilides A–J (**151**–**160**) and xeniaethers C–E (**161**–**163**), from *Xenia* sp. collected at Bonotsu, Kagoshima Prefecture, Japan [22,54]. In addition, azamilides A–J (**151**–**160**) were the first example of xenicane-type diterpenoids from the genus *Xenia* with an open A-ring. In 1999, three new diterpenoids, namely xeniaol (**164**), xeniadiol (**165**), and xeniatriol (**166**), were isolated from *Xenia* sp. collected at Ishigaki Island, Okinawa, Japan [55]. In 2000, one new diterpenoid, xeniaoxolane (**167**), was isolated from *Xenia* sp. collected at Ishigaki Island, Okinawa, Japan [56]. In 2002, two new xeniolides, xeniolide-F (**168**) and 9-hydroxyxeniolide-F (**169**), were isolated from *Xenia* sp. collected at Togian Islands near Sulawesi Island, Indonesia [57]. In 2003, two new diterpenoids, dihydroxeniolide-A (**170**) and isoxeniatriacetate (**171**), were isolated from *Xenia* sp. collected at Ishigaki Island, Okinawa, Japan [58]. In 2008, four new diterpenoids, xenimanadins A–D (**172**–**175**), were isolated from *Xenia* sp. collected at Bunaken Marine Park off Manado, North Sulawesi, Indonesia [59]. In *2015*, three new diterpenoids (**176**–**178**) without given names were isolated from *Xenia* sp. collected at Miyako Island, Okinawa, Japan [60]. From 2017 to 2018, two new diterpenoids, 12-epi-9-deacetoxyxenicin (**179**) and 15-deoxy-isoxeniolide-A (**180**), were isolated from *Xenia* sp. collected at Mengalum and Mantanani Islands, Sabah, Malaysia [5,61]. In addition, compound **179** showed cytotoxicity against adult T-cell leukemia S1T cells at IC_50_ 5.89 µg/mL (Figure 8) [61].

## 3. Sesquiterpenes

The first sesquiterpenes isolated from soft coral *Xenia* were palustrol (**181**) and 7-acetoxymuurolene (**182**) from *X*. *obscuronata* collected at the Gulf of Eilat, the Red Sea [37]. In 1986, three new sesquiterpenoids, namely 1,2-epoxy-4-isopropyl-1,6-dimethyl-1,2,3,4,4a,7,8,8a-octahydronaphthalene (**183**), 4-isopropyl-6-methyl-3,4,4a,7,8,8a-hexahydronaphthalene-1-methanol (**184**), and 5,6-epoxy-4-isopropyl-6-methyl-3,4,4a,5,6,7,8,8a-octahydronaphthalene-1-methanol acetate (**185**), were isolated from two species of *Xenia* collected at Rib Reef, in the Central Region of the Great Barrier Reef north of Townsville and Hook Reef near the Whitsunday Islands south of Townsville [62]. In 1987, one new sesquiterpenoid, 3,3,7,11-tetramethyltricyclo[6.3.0.0]undec-1(8)-en-4-ol (**186**), isolated from *X. novae-britanniae,* was collected at Rib Reef, Townsville, Australia [63]. In 2002, six new cadinenes, xenitorins A–F (**187**–**192**), were isolated from *X. puerto-galerae* collected at Green Island, off Taiwan [64]. In 2007, three new cadinenes, namely 8-*epi*-xenitorin A (**193**), 10-*epi*-xenitorin C (**194**), and 7-isopropenyl-4,10-dimethyl-2,3,4,5-tetrahydronaphthalene (**195**), were isolated from *X. puerto-galerae* collected at Green Island, off Taiwan [65]. In 2018, one new sesquiterpenoid, 10β-*O*-methyl-1αH,5αH-guaia-6-en-4β-ol (**196**), was isolated from *X. stellifera* collected at Mengalum Island, Sabah, Malaysia [6] (Figure 9).

## 4. Steroids

In 1986, four new steroids, xeniasterols a–d (**197**–**200**), were isolated from *Xenia* sp. collected at Zamami-jima, Okinawa Prefecture, Japan [66]. Moreover, steroids **197**–**200** showed growth inhibitory activity against B-16 melanoma cells at IC_50_ 5 µg/mL. In 2016, one new steroid, pentahydroygorgosterol (**201**), was isolated from *X. umbellata* collected at Red Sea Coast, Jeddah, Saudi Arabia [48] (Figure 10).

## 5. Synthesis of *Xenia* Diterpenoids

As reviewed in the earlier part of this paper, *Xenia* is well known as a rich source of bioactive compounds, especially possessing anticancer activity. The promising anticancer and antimicrobial activities of the isolated compounds from *Xenia* as lead pharmaceuticals have attracted the interest of synthetic chemists. However, synthesis of the diterpenoids is challenging, especially when they consist of an *E*-configuration on the nine-membered rings. Various synthetic approaches have been proposed for a few total syntheses of xeniolides. Ring expansion, ring-closing, and ring-contracting reactions are among the approaches depending on the individual substitution pattern of the target compound [67]. As for reports on the synthesis of such diterpenoids, there is an apparent lack of data due to the challenge of synthesizing nine-membered rings. To date, there is no reported total synthesis of xenicins, and only a few total syntheses of xeniolides have been accomplished [67].

## 6. Conclusions

This paper presents a review of the terpenes discovered in the soft coral *Xenia* and their biological activities. From 1977 to 2018, a total of 199 terpenes, including 14 sesquiterpenes, 180 diterpenes and 5 steroids, were found in soft coral *Xenia* for the first time. These species of soft coral *Xenia,* including *X. blumi*, *X. crassa*, *X. elongata*, *X. faraunensis*, *X. florida*, *X. lilielae*, *X. macrospiculata*, *X. membranacea*, *X. novae-britanniae*, *X. obscuronata*, *X. puerto-galerae*, *X. stellifera*, *X. umbellata*, *X. viridis,* and *Xenia* sp., were primarily collected at Green Island in Taiwan, regions of Kagoshima and Okinawa in Japan, Gulf of Eilat at the Red Sea, Townsville in Australia, and Sarcelle Pass south of New Caledonia. The assessment of cytotoxicity against cancer cell lines is the most frequent biological evaluation reported for these terpenes, and most of them were not widely tested for their biological potential. An investigation of the diversity in soft coral family Xeniidae revealed that *Xenia* was polyphyletic [68]. This suggested that it might not be suitable to place soft coral belonging to the genus *Xenia* in a single taxon as their morphological characteristics used for traditional taxonomy are not well reflected in the phylogeny in this genus [68]. A recent study from Maloney and McFadden’s group suggested that chemical studies in the last 50 years on soft coral *Sarcophyton* may have unknowingly comprised different cryptic species of *Sarcophyton glaucum*, leading to apparently distinctive chemical variation [69]. In the future, such studies should be carried out for soft coral *Xenia* to gain more understanding of their idiosyncratic chemical variation. In the course of this review, we realized that most of the early reports of natural products chemistry of this genus did not place much importance on the proper authentication of the collected specimens. Most did not report voucher specimen numbers or deposits at biodiversity collection centers. This is quite understandable, but with the Nagoya Protocol coming into force in 2014, these are important matters that a natural products chemist can no longer avoid. Therefore, because the collected marine invertebrate specimens always exhibit a high degree of morphological plasticity and differences due to geographical locations, their proper identification and voucher specimen deposition is warranted for future reference. In addition, recent advancement of technology, particularly in DNA sequencing, has provided sufficient molecular information to facilitate soft coral identification in a more precise and swift manner. The authors recommend the inclusion of morphological and molecular information in articles in future studies. It is also important to realize that compounds isolated from these invertebrates are important not only for their biological potential, but also as a vital tool for chemo-taxonomical studies.

## Figures and Tables

**Figure 1 molecules-25-05386-f001:**
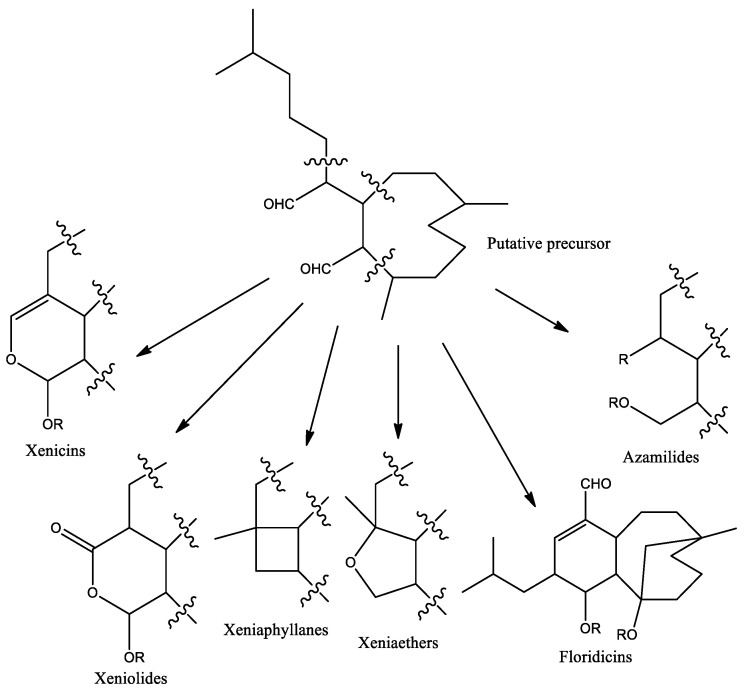
Diversity of chemical skeletons discovered from soft coral genus *Xenia*.

**Figure 2 molecules-25-05386-f002:**
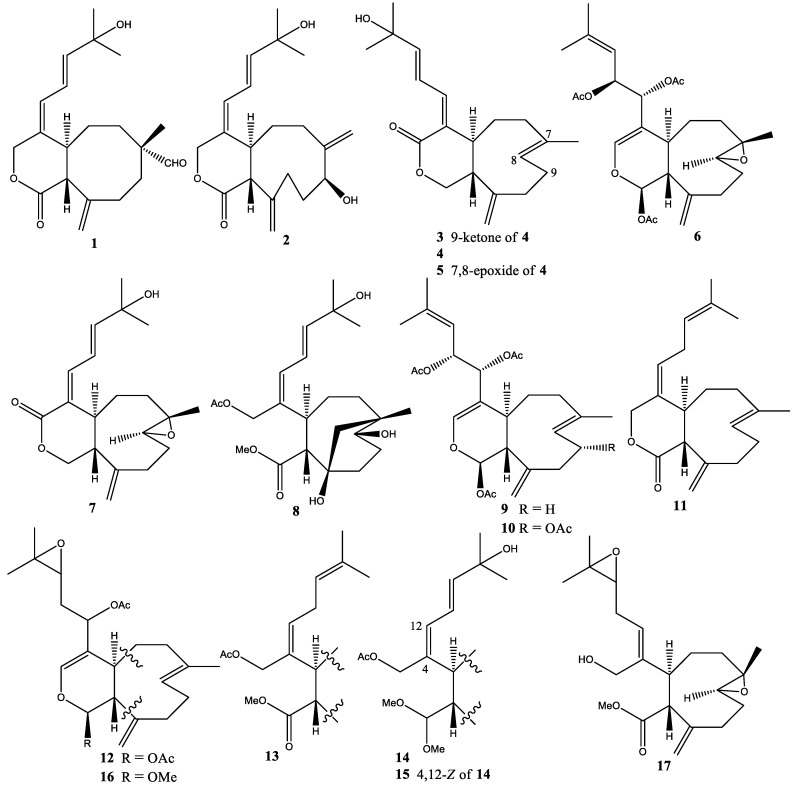
Structures of compounds **1–17**.

**Figure 3 molecules-25-05386-f003:**
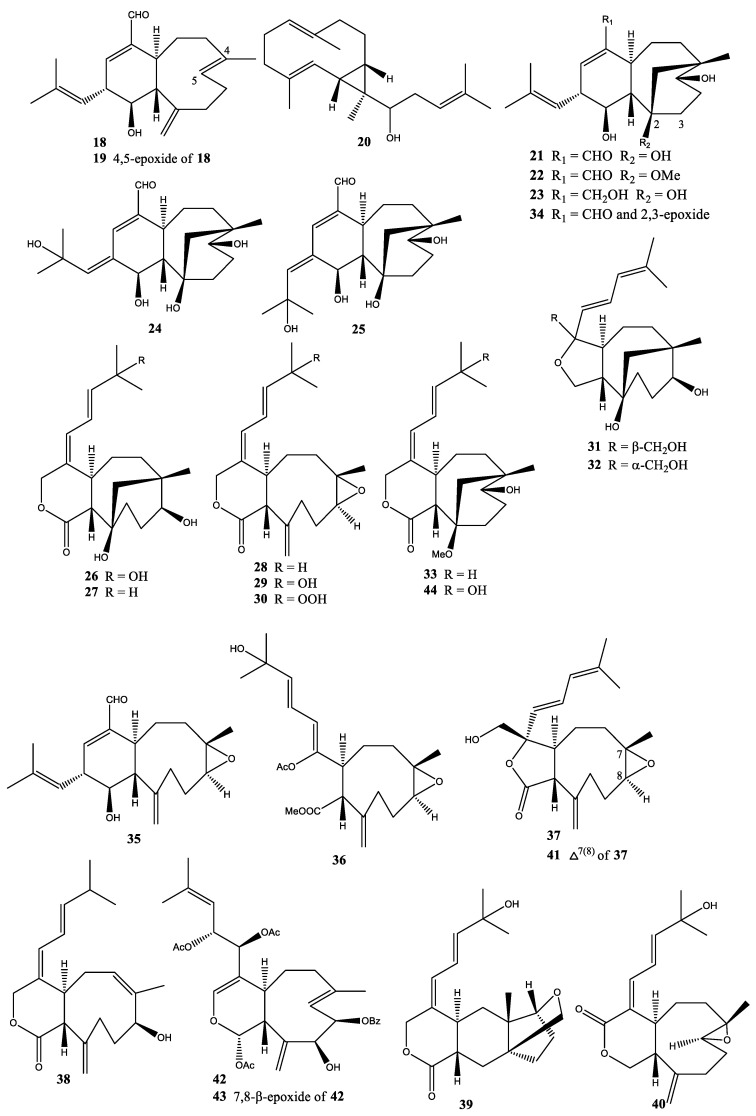
Structures of compounds **18–44**.

**Figure 4 molecules-25-05386-f004:**
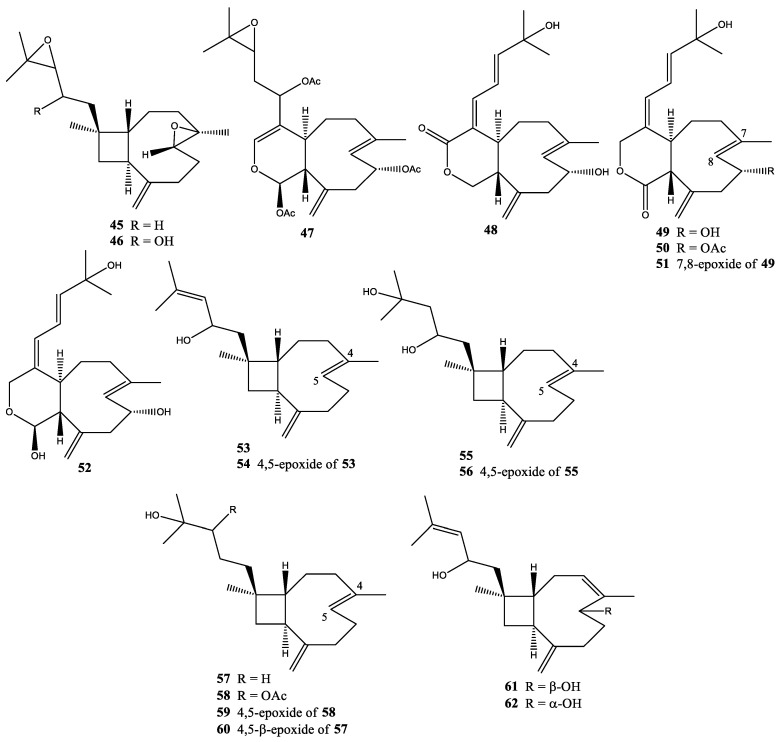
Structures of compounds **45**–**62**.

**Figure 5 molecules-25-05386-f005:**
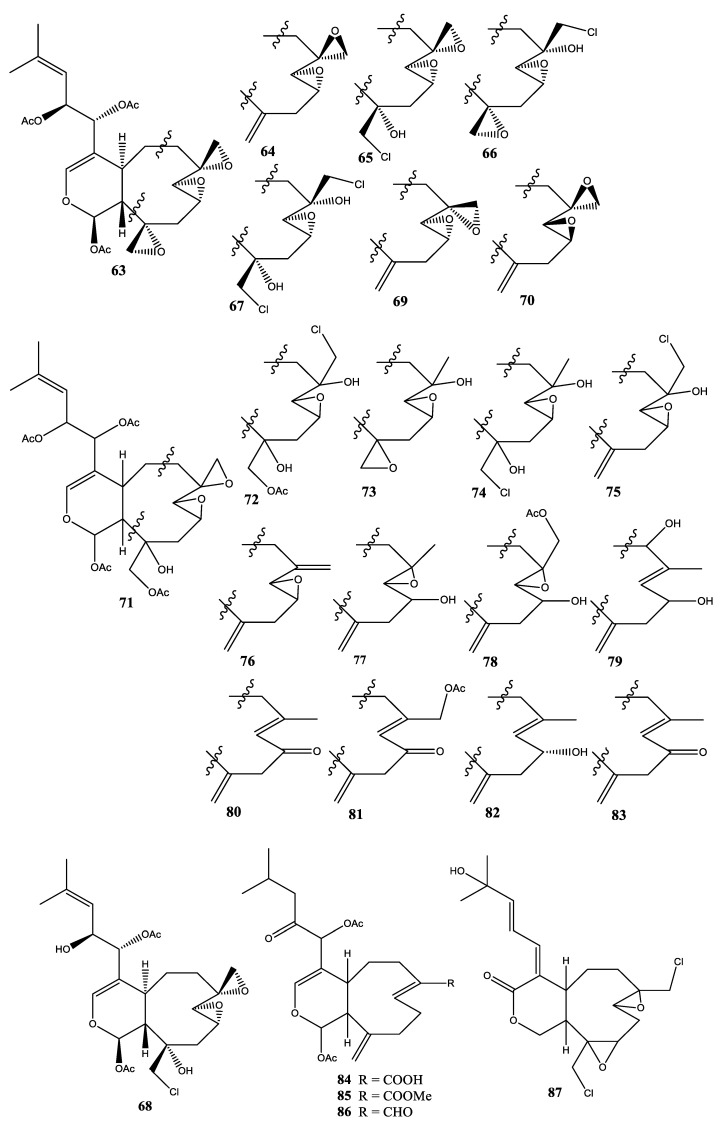
Structures of compounds **63**–**87**.

**Figure 6 molecules-25-05386-f006:**
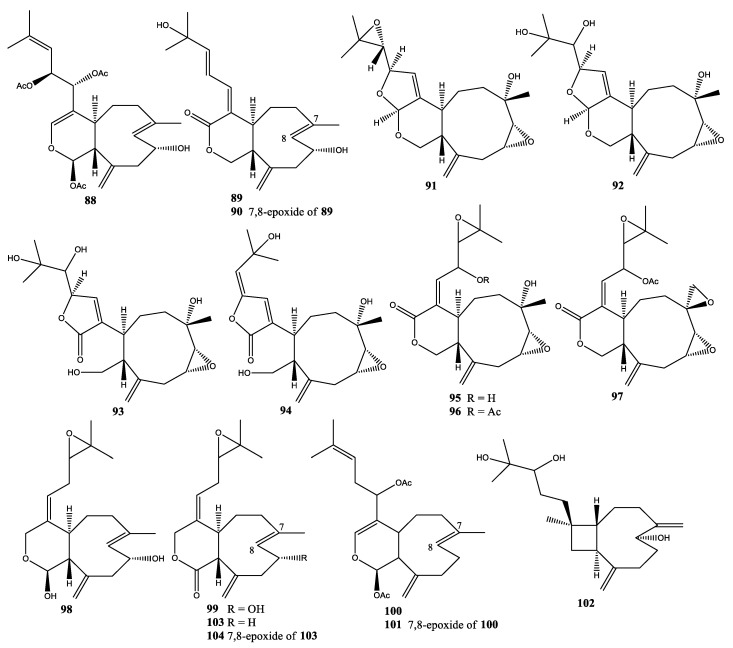
Structures of compounds **88**–**113** and **123**,**124.**

**Figure 7 molecules-25-05386-f007:**
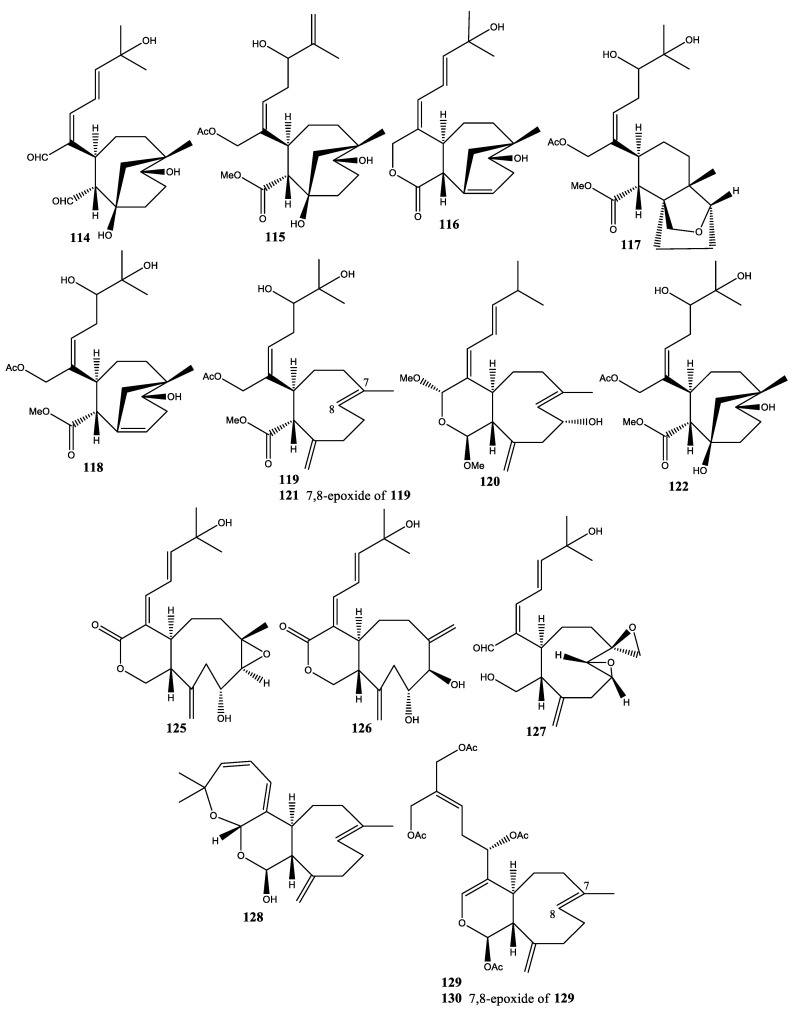
Structures of compounds **114**–**122** and **125**–**130**.

**Figure 8 molecules-25-05386-f008:**
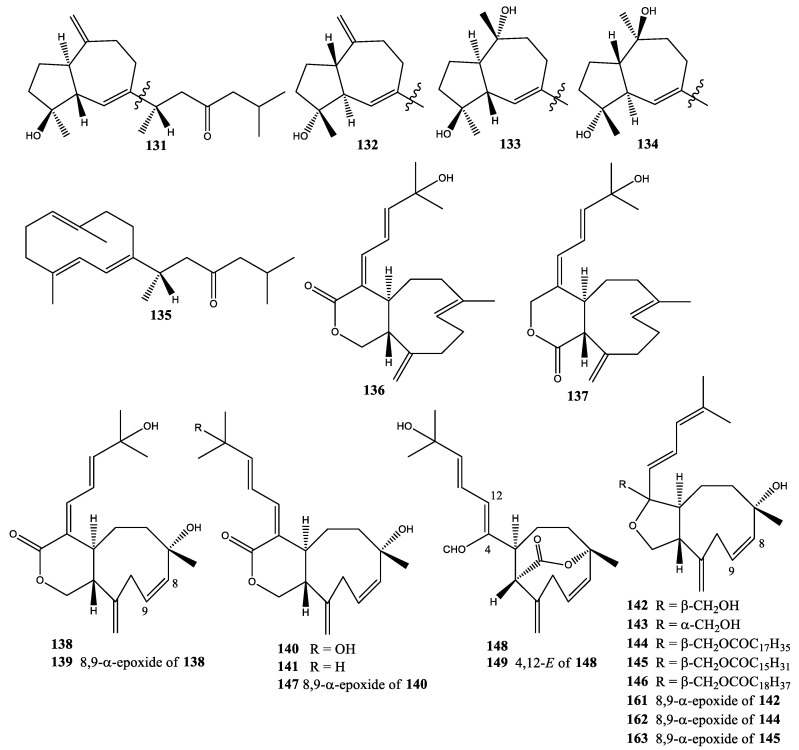
Structures of compounds **131–180**.

**Figure 9 molecules-25-05386-f009:**
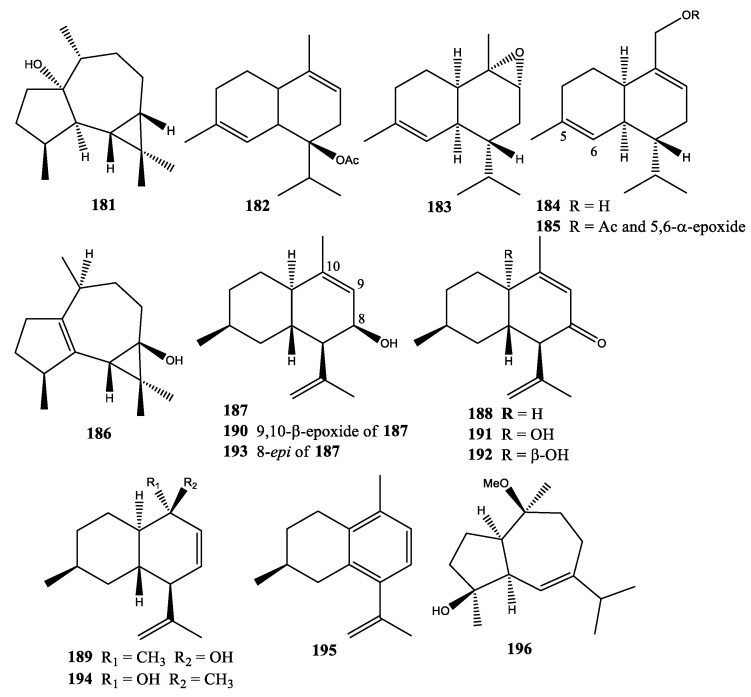
Structures of compounds **181**–**196**.

**Figure 10 molecules-25-05386-f010:**
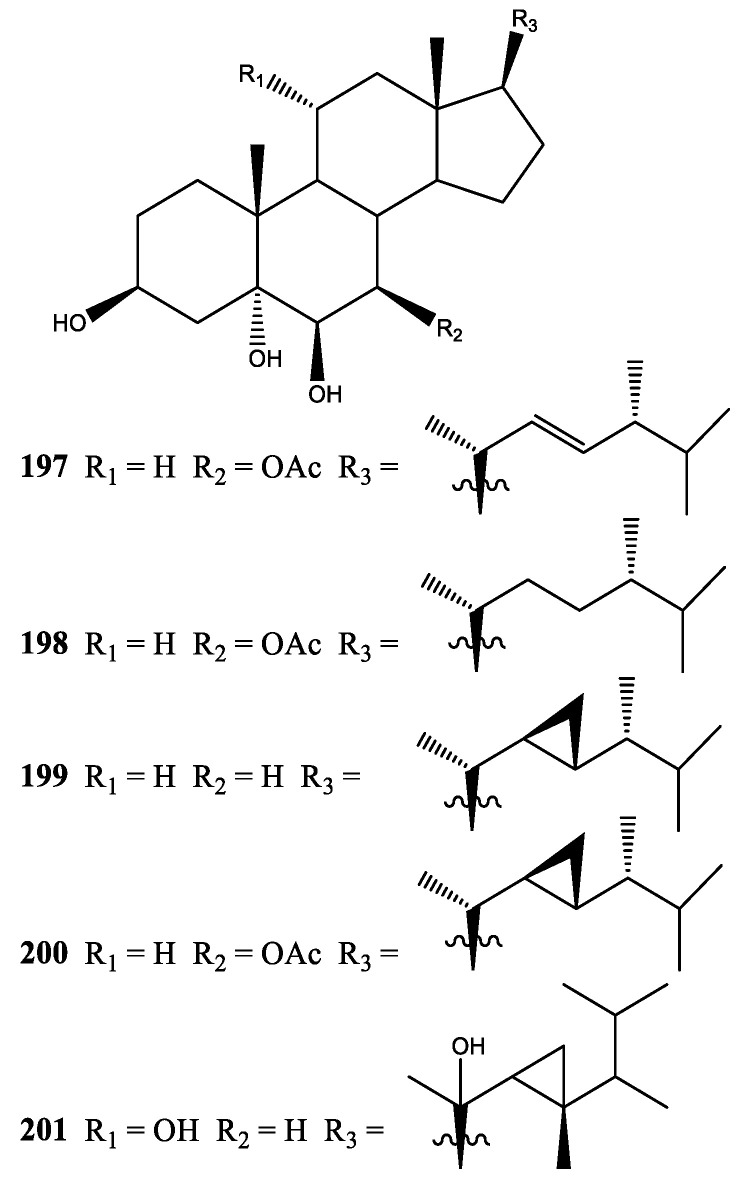
Structures of compounds **197**–**201**.

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
