# Peer review of "Terpenoids from Marine Soft Coral of the Genus Xenia in 1977 to 2019"

_molecules, 2020, doi:10.3390/molecules25225386_

Round 1
Reviewer 1 Report
The review article is an overview of terpenoids isolated from marine soft coral Xenia. The authors presented in their article the chemical diversity of terpenoids with chemical structures displayed in the terpenoid subcategories. The location of the Xenia specimens along with the biological activity of the isolated compounds were outlined. Overall, the review is clearly presented and structured. I have a few comments for the authors regarding their review, which are presented below:
Since the authors emphasized in the abstract that the review is an overview of the biological activity of terpenoids from Xenia, this information needs to be systematized. It would be informative for the reader of this review to summarize the biological activity of the terpenoids in the form of a Table. I suggest summarizing the cytotoxic activity of Xenia terpenoids, since the authors mention that it is the most frequently provided information on the activity of these corals.
I suggest the authors unify the provided information regarding the cytotoxic activity of compounds by adding information, where omitted, regarding the names of the cell lines or their tissues of origin. For example in line 61, please add the following information: HT-29 (colorectal cancer) and P-388 (murine leukemia) cell lines.
The manuscript requires moderate language corrections. Some examples are presented below:
Line 25: attracted the attention
Line 37: literature
Line 46: in the 21st century
Line 73, 76, 225: given names
Line 88: followed by the isolation
Line 76: X. elongate, purchased
Line 87: This diterpenoid was considered the first possessing
Line 96: were isolated from X. florida
Line 108: One year after the discovery of xenicin, a second
Line 119: by the same
Line 145: has remained
Line 180: MIC values of
Line 203: were isolated from Xenia sp.
Line 206: showed the greatest antibacterial potency
Line 233: the Gulf
Line 269: especially possessing anticancer activity
Line 275: As for reports on the synthesis of such diterpenoids, there is an apparent lack of data due to the challenging synthesis of nine-membered rings
Line 277: there is no reported total synthesis of xenicins, only
288: biologically evaluated
290: which belong to the genus Xenia might not be suitable to place in a single taxon, since
291: reflected in the phylogeny of this genus
292: suggested that the past
294: In the future, such studies
Author Response
RESPONSE TO REVIEWER 1 COMMENTS
The review article is an overview of terpenoids isolated from marine soft coral Xenia. The authors presented in their article the chemical diversity of terpenoids with chemical structures displayed in the terpenoid subcategories. The location of the Xenia specimens along with the biological activity of the isolated compounds were outlined. Overall, the review is clearly presented and structured. I have a few comments for the authors regarding their review, which are presented below:
Q-1: Since the authors emphasized in the abstract that the review is an overview of the biological activity of terpenoids from Xenia, this information needs to be systematized. It would be informative for the reader of this review to summarize the biological activity of the terpenoids in the form of a Table. I suggest summarizing the cytotoxic activity of Xenia terpenoids, since the authors mention that it is the most frequently provided information on the activity of these corals.
R-1: We appreciate the reviewer suggestion but we the authors have decided to retain the current arrangement format of the biological activity information.
Q-2: I suggest the authors unify the provided information regarding the cytotoxic activity of compounds by adding information, where omitted, regarding the names of the cell lines or their tissues of origin. For example in line 61, please add the following information: HT-29 (colorectal cancer) and P-388 (murine leukemia) cell lines.
R-2: Yes. It has been improved.
Q-3: The manuscript requires moderate language corrections. Some examples are presented below:
Line 25: attracted the attention
Line 37: literature
Line 46: in the 21st century
Line 73, 76, 225: given names
Line 88: followed by the isolation
Line 76: X. elongate, purchased
Line 87: This diterpenoid was considered the first possessing
Line 96: were isolated from X. florida
Line 108: One year after the discovery of xenicin, a second
Line 119: by the same
Line 145: has remained
Line 180: MIC values of
Line 203: were isolated from Xenia sp.
Line 206: showed the greatest antibacterial potency
Line 233: the Gulf
Line 269: especially possessing anticancer activity
Line 275: As for reports on the synthesis of such diterpenoids, there is an apparent lack of data due to the challenging synthesis of nine-membered rings
Line 277: there is no reported total synthesis of xenicins, only
288: biologically evaluated
290: which belong to the genus Xenia might not be suitable to place in a single taxon, since
291: reflected in the phylogeny of this genus
292: suggested that the past
294: In the future, such studies
R-3: Yes. All the language corrections have been improved.

Reviewer 2 Report
General comments:
Authors reported some information related with terpenoids from marine soft coral of the genus Xenia and provide interesting chemical structures of these compounds. However, authors just compile information related with the date and the source of isolation, and in some cases the bioactivity of terpenoids (they reported 8 versus the 199 contained in in soft coral Xenia). They did not provide the concentration of each isolated terpenoid, neither support their revision with a justification, purpose or contribution. There is no author´s point of view, criticism and recommendations.
Specific comments:
The abstract must be extended and improved for providing a wide context related with this review. Additionally, the purpose of the review must be added. What is the contribution of this review?
Introduction: It must be extended and improved for providing a wide context related with this review. Please, also provide a justification of this review.
Line 38: The title of this section is “Diterpenes” but the information was only focused on a small group, the xenicane diterpenes. Authors should add a general information of diterpenes, a classification and latter talk about xenicane.
It will be interesting to provide the concentration of each isolated terpenoid.
Why authors only provided the “bioactivity” of 8 compounds if there are around 199 (according to the conclusion)? In addition, information related with the method of extraction could be interesting to take into consideration.
Lines 272-273: Why this topic (synthesis of xeniolides) was not considered in this revision?
Author Response
RESPONSE TO REVIEWER 2 COMMENTS
General Comments: Authors reported some information related with terpenoids from marine soft coral of the genus Xenia and provide interesting chemical structures of these compounds. However, authors just compile information related with the date and the source of isolation, and in some cases the bioactivity of terpenoids (they reported 8 versus the 199 contained in in soft coral Xenia). They did not provide the concentration of each isolated terpenoid, neither support their revision with a justification, purpose or contribution. There is no author´s point of view, criticism and recommendations.
Specific comments:
Q-1: The abstract must be extended and improved for providing a wide context related with this review. Additionally, the purpose of the review must be added. What is the contribution of this review?
R-1: Yes, it has been improved.
Q-2: Introduction: It must be extended and improved for providing a wide context related with this review. Please, also provide a justification of this review
R-2: Yes, it has been improved.
Q-3: Line 38: The title of this section is “Diterpenes” but the information was only focused on a small group, the xenicane diterpenes. Authors should add a general information of diterpenes, a classification and latter talk about xenicane.
R-3: Yes, it has been improved.
Q-4: It will be interesting to provide the concentration of each isolated terpenoid.
R-4: The authors have decided to remain the current format of the description of isolated terpenoid. But we appreciate the reviewer suggestion.
Q-5: Why authors only provided the “bioactivity” of 8 compounds if there are around 199 (according to the conclusion)? In addition, information related with the method of extraction could be interesting to take into consideration.
R-5: We have provided more than 8 compounds that have exhibited potent activities from all that was reviewed in our paper. We have also look into the extraction approach in all the papers reviewed, it is relatively similar. All sample extractions involved the traditional technique, where the chopped specimens were soaked organic solvents such as methanol or ethanol. Only a few used acetone, ethyl acetate and chloroform. However, we appreciate the reviewer suggestion.
Q-6: Lines 272-273: Why this topic (synthesis of xeniolides) was not considered in this revision?
R-6: The explanation has been included in lines 290-301.

Round 2
Reviewer 1 Report
I have the following comments for the authors following their revision:
- The addition of information regarding the names of the cell lines or their tissues of origin in some cases was over-looked. The following sentences require the addition of information regarding either tissue source or cell line:
Line 110: human liver carcinoma cells - please add cell line
Line 182: P-388 cell line - please add tissue source
Line 183: A-549, HT-29 and P-388 cell lines - please specify tissue sources
Line 166: mammalian cells – please specify
- Please add literature references for information provided in the following sentences:
Line 70-75
Line 90-91
Line 95-96
Line 109-110
Line 165-168
Line 242- 243
- The inserted fragments require language corrections.
Author Response
Response to the comments of Reviewer-1 is attached below.

Reviewer 2 Report
Authors have considerably improved the manuscript, mainly in sections such as the abstract, introduction and content. However, some important comments were omitted:
- The main topic of the manuscript is “terpenoids from marine soft coral”… in my pervious comment I suggested to provide the concentration of isolated terpenoids but authors decided skipping this comment. In my oppinion, this suggestion was one of the most important.
- There is a lack of important items for enhancing this kind of publication such as the authors’ point of view, criticism and recommendations. Authors should take the challenge and provide information on this regard.
Author Response
Response to the comments of Reviewer-2 is attached below.

Round 3
Reviewer 2 Report
Authors have provided an improvement on the manuscript. They have taken the challenge for adding their point of view, criticism and recommendations, that is great!
I understood their point of view after reading the response in relation to terpenoids concentration and I agree. They have added some information on this regard in the manuscript, which is important for readers’ comprehension.
Authors must improve the English language on the added sections. Afterwards, the manuscript can be considered for publication on Molecules.